# Insight into Unusual Supramolecular Self-Assemblies of Terthiophenes Directed by Weak Hydrogen Bonding

**DOI:** 10.3390/ijms241311127

**Published:** 2023-07-05

**Authors:** Shiv Kumar, Kristof Van Hecke, Franck Meyer

**Affiliations:** 1Microbiology, Bioorganic and Macromolecular Chemistry (MBMC) Unit, Faculty of Pharmacy, Université Libre de Bruxelles, 1050 Brussels, Belgium; shiv.kumar@ulb.be; 2XStruct, Department of Chemistry, Ghent University, Krijgslaan 281-S3, 9000 Ghent, Belgium; kristof.vanhecke@ugent.be

**Keywords:** terthiophene, bipyridine, hydrogen bonding, crystal structure, Hirshfeld surface, cocrystal, fluorescence

## Abstract

A supramolecular self-assembly of semiconducting polymers and small molecules plays an important role in charge transportation, performance, and lifetime of an optoelectronic device. Tremendous efforts have been put into the strategies to self-organize these materials. In this regard, here, we present the self-organization of terthiophene and its methyl alcohol derivative with 4,4′-bipyridine (44BiPy). An unexpected 2D layered organization of 5,5″-dimethyl-2,2′:5′,2″-terthiophene (DM3T) and 44BiPy was obtained and analyzed. Single-crystal X-ray diffraction analysis revealed that DM3T and 44BiPy consist of stacked, almost independent, infinite 2D layers while held together by weak hydrogen bonds. In addition to this peculiar supramolecular arrangement of these compounds, the investigation of their photophysical properties showed strong fluorescence quenching of DM3T by 44BiPy in the solid state, suggesting an efficient charge transfer. On the other hand, the methyl alcohol derivative of terthiophene, DM3TMeOH, organized in a closed cyclic motif with 44BiPy via hydrogen bonds.

## 1. Introduction

Over the last several decades, organic π-conjugated compounds have emerged as a new class of semiconducting materials because of their fascinating optical and electronic properties offering a wide range of applications in organic light-emitting diodes (OLEDs), organic solar cells (OSCs), organic field-effect transistors (OFETs), organic lasers, optical or electrochemical sensors, and many more [1,2,3,4,5,6,7]. Thiophene and its derivatives, alone or along with many other aromatic and heteroaromatic constituents, are well-known building blocks in the construction and development of today’s high-performance organic semiconducting materials [8]. Thousands of papers on thiophene-based compounds have been published over the last several decades describing the rational design, synthetic methodology, and investigation of their semiconducting and photophysical properties. Thiophene oligomers (e.g., bithiophene, terthiophene, and quaterthiophene) and polymers (e.g., poly(3-hexylthiophene) (**P3HT**) and poly(3,4-ethylenedioxythiophene):polystyrene sulfonate (**PEDOT:PSS**)) are widely studied and used materials in the fabrication of organic optoelectronic devices [9].

As we know, without any substituent, polythiophene is a poorly soluble polymer in most common organic solvents, which makes it difficult to process for fundamental study of charge transportation and redundant to use for any practical application. Therefore, for practical purposes, solution-processed poly(3,(4)-(di)alkyl)thiophenes have been obtained by introducing alkyl chains at the backbone polythiophenes and utilized various optoelectronic application. Nevertheless, many well-defined oligothiophenes have been synthesized and studied to serve as the basis for a thorough understanding of the electrical properties of polythiophenes. Among these oligothiophenes, terthiophene and its derivatives, because of their facile synthesis, are widely studied to understand the electronic structure and charge transport mechanism in this class of materials. For example, Birnbaum and Kohler recorded the fluorescence excitation and emission spectra of 2,2′:5′,2″-terthiophene and performed a detailed analysis of the lowest-energy excited singlet state [10], which was followed by the publication of hundreds of papers, investigating theoretical and experimental photophysical and electrochemical properties to establish a structure–property relationship in terthiophenes and oligothiophenes [11,12,13,14,15,16,17]. Pappenfus and coworkers synthesized a terthiophene-based quinodimethane derivative and fabricated a thin-film field-effect transistor to measure the charge mobility [14,15]. These studies have also immensely helped to understand and extrapolate the photophysical, thermal, electrochemical, and charge transport properties of polythiophenes, albeit theoretically. Furthermore, the self-assembly of organic semiconductors plays an important role in electronic transportation, thus acting as a deciding factor in the performance of an organic optoelectronic device [18,19].

The synthesis, purification, and characterization of alkyl-substituted and/or alkyl-capped oligothiophene have been reported in literature [20,21]. However, the crystal structure of methyl-capped terthiophene has been missing. Herein we describe the synthesis, single-crystal X-ray structure, and crystal packing of 5,5”-dimethyl-2,2′:5′,2″-terthiophene (**DM3T**) (Figure 1). Weak hydrogen bonding-driven supramolecular self-assembly of **DM3T** was studied, and Hirshfeld surface analysis was carried out to support the interactions. Furthermore, in our recently published study, the supramolecular self-assemblies of thiophene derivatives via halogen bonding [22] led to the observation of a N···S interaction between the S-atom of a thiophene derivative and N-atom of 4,4′-bipyridine (**44BiPy**) [23,24]. To verify the universal viability of the N···S interaction, we co-crystallized **DM3T** with 4,4′-bipyridine (**44BiPy**) via slow evaporation of the solvent, hoping that these two molecules would organize into a supramolecular self-assembly via chalcogen bonding (ChB). However, this experiment led to a totally unexpected result. 

The crystal structure and Hirshfeld surface analysis of **DM3T–44BiPy** is presented. We also present the synthesis of (5,5′′-dimethyl-[2,2′:5′,2″-terthiophen]-3′-yl)methanol (**DM3TMeOH**) and its hydrogen-bonded cocrystal structure analysis with **44BiPy**. A detailed comparison of structural analysis of **DM3T**, **DM3T–44BiPY**, and **DM3TMeOH–44BiPy** is presented. We then discuss the photophysical properties and fluorescence quenching of **DM3T** and **DM3TMeOH** in the presence of **44BiPy**. The chemical structures of the **DM3T**, **DM3TMeOH**, and **44BiPy** are shown in Figure 1.

## 2. Results

### 2.1. Synthesis

**DM3T** was prepared by a palladium-catalyzed Suzuki cross-coupling reaction [25] between 2,5-dibromothiophene and 5-methylthiophene-2-boronic acid pinacol ester in the presence of tetrakis(triphenylphosphine)palladium (Pd(PPh_3_)_4_) as a catalyst derivative (Appendix A). On the other hand, **DM3TMeOH** was prepared starting from 3-thiophenemethanol (**TMeOH)** via a multistep organic synthesis procedure as depicted in Figure 1. Briefly, the bromination of **TMeOH** with NBS in dichloromethane at room temperature gave a dibromo derivative (**DBTMeOH**) which was subsequently subjected to protection of –OH group by *tert*-butyldimethylsilyl chloride (TBDMSCl). Following the same Suzuki cross-coupling reaction conditions and reagents as used to synthesize **DM3T** except using newly synthesized **DBTMeOTBDMS** instead of 2,5-dibromothiophene gave the –OH-protected desired terthiophene derivative, **DM3TMeOTBDMS**, which was easily deprotected by treatment with TBAF in THF at room temperature to yield **DBTMeOH** (Appendix A).

### 2.2. Crystal Growth

Crystal growth was conducted via slow evaporation of a mixture of solvents. Bright-green-yellow single crystals of **DM3T** were obtained from slow evaporation of a dichloromethane and hexane solution. Similarly, pale-yellow needle-shaped cocrystals were obtained from a solution consisting of **DM3T** (2 equiv.) and **44BiPy** (3 equiv.), while pale-green cocrystals were obtained for **DM3TMeOH–44BiPy** from a solution of **DM3TMeOH** (2 equiv.) and **44BiPy** (1 equiv.). All single crystals were subsequently analyzed.

### 2.3. Single-Crystal X-ray Diffraction Analysis

#### 2.3.1. DM3T

Single-crystal X-ray diffraction analysis revealed the characteristic conformational parameters of **DM3T**. **DM3T** crystallized in the monoclinic non-centrosymmetric space group *P*2_1_ with four molecules per unit cell. All the thiophene units adopted an S-*trans* conformation with slightly curved molecular plane geometry (Figure 2). The asymmetric unit was composed of two **DM3T** molecules, held together by strong C–H···π interactions. These molecules aligned themselves at an angle of 57.95° with respect to their molecular planes. To form the single crystal, the neighboring molecules organized themselves longitudinally and transversally via several noncovalent interactions. The metrics of all the noncovalent interactions present in the **DM3T** single crystal are summarized in Table 1.

In the transverse arrangement of **DM3T** molecules, several noncovalent interactions were observed. In addition to the intermolecular C21–H21···S1 and intramolecular C11–H11···S2 hydrogen bonds, there were strong C–H···π interactions involving C3–H3·· π, C8–H8···π, C24–H24···π, and C28–H28···π. Furthermore, weak chalcogen bonds (ChBs) were evidenced by the S23···π interaction (d = 3.868 Å), as well as chalcogen–chalcogen contacts between S1 and S21 (d = 3.690(3) Å). Although the distances between these noncovalent interactions were slightly higher than the sum of their van der Waals radii [26], their presence was supported by Hirshfeld surface analysis. The supramolecular packing exhibited a herringbone pattern.

The relative strengths and contributions of intermolecular interactions of a crystalline compound can be visualized by Hirshfeld surface analysis. For this purpose, a software program, CrystalExplorer (Version 21.5, Revision: 608bb32), was used to generate and analyze the Hirshfeld surface (Figure 3), shape index, and curvedness surfaces (Appendix A). In our study, we conducted an in-depth analysis of the crystal packing of **DM3T** using Hirshfeld surface analysis and fingerprint plots. Figure 4 presents the overall and resolved 2D fingerprint plots of **DM3T**, which provide valuable insights into the atom pair interactions and their relative contributions to the Hirshfeld surface.

The fingerprint plots revealed that the major contributions to the Hirshfeld surface of **DM3T** arose from three types of interactions: H···H (40.2%), C···H/H···C (29.7%), and S···H/H···S (18.7%). These interactions involved hydrogen bonding, as well as C–H···π interactions between carbon and hydrogen atoms. The significant contribution from H···H interactions suggests that the vast majority of London dispersion forces were acting within the crystal structure. Additionally, the fingerprint plots showed lower proportions of S···C/C···S (7.8%) and S···S (3.6%) interactions. The presence of S···C/C···S interactions indicates the presence of chalcogen bonds, while S···S interactions represent the chalcogen–chalcogen contacts, and both contribute to the overall crystal packing of **DM3T**. The Hirshfeld surface analysis, combined with the fingerprint plots, provides a comprehensive understanding of the atom pair interactions such as HB, CH···π, ChB, and S···S contacts and highlights their significance in the arrangement of **DM3T** molecules within the crystal lattice.

#### 2.3.2. DM3T–44BiPy

Single-crystal X-ray diffraction analysis of the cocrystal agreed well with the expected structure of the constituents, **DM3T** and **44BiPy**, as shown in Figure 5. The cocrystal belonged to the triclinic space group *P*$\bar{1}$ with five molecules in the asymmetric unit, two **DM3T** and three **44BiPy** molecules. In **DM3T**, all the thiophene units were in S-*trans* conformation. These molecules were held together by strong HB, C–H···π, and S···S interactions.

The typical distances of C–H···π interactions were within the range of 2.714–3.094 Å, while the S6···S6 separation was measured at 3.554 Å, which is approximately 1.3% shorter than the sum of van der Waals radii of sulfur atoms (Figure 5). Even stronger S···S short contacts, with separation measured at 3.491 Å, were observed and reported for 2,2′:5′,2′′-terthiophene-5-carbaldehyde [27]. Appendix A shows the complex network of HBs involving C–H···N interactions in cohesion of **44BiPy** molecules. The details of all noncovalent interactions are summarized in Table 2. Crystal data, data collection, and structure refinement details are summarized in Appendix A.

The supramolecular cocrystal packing of **DM3T–44BiPy** revealed a remarkable and distinctive layered structure, as depicted in Figure 6. Both **DM3T** and **44BiPy** compounds self-organized into independent infinite layers, which were stacked on top of one another and connected by weak hydrogen-bonding interactions. The **DM3T** layer was primarily stabilized by a combination of hydrogen bonding and short contacts between sulfur atoms (S···S). In contrast, the **44BiPy** molecules formed an infinite layer characterized by a complex arrangement, primarily stabilized by strong HBs.

Figure 6 illustrates that the average distance between any two layers of **DM3T** and **44BiPy** was measured to be 15.291 Å and 14.076 Å, respectively. These data highlight the separation between adjacent layers in the crystal structure.

The observed layered arrangement and intermolecular interactions in the supramolecular cocrystal packing of **DM3T–44BiPy** are noteworthy. The distinct characteristics of the individual layers and their stacking provide valuable insights into the crystal packing structure. Understanding these arrangements is crucial for comprehending the properties and potential applications of this cocrystal system (see Section 2.4).

Figure 7 presents the Hirshfeld surface of **DM3T–44BiPy**, visualized by mapping over *d*_norm_ within the range of −0.05 to 1.30 Å. The surface was prominently filled with red spots, indicating regions where the intermolecular contacts were shorter than the van der Waals separation. These spots were attributed to a complex network of hydrogen bonds, as observed in the X-ray structure of **DM3T–44BiPy** (Appendix A). The formation of these hydrogen bonds contributed to the overall stability and organization of the crystal structure. Additionally, the shape index and curvedness surfaces, which provide further insights into the molecular arrangement, were generated (Appendix A).

Figure 8 depicts the overall and resolved 2D fingerprint plots, which illustrate the atom pair interactions and their respective contributions to the Hirshfeld surface of **DM3T–44BiPy**. As anticipated, the major contributions stemmed from various types of hydrogen-bonding interactions, occurring simultaneously within the crystal structure. The relative contributions of these interactions were as follows: H···H (41.5%), C···H/H···C (28.9%), N···H/H···N (12.9%), and S···H/H···S (9.9%). While the proportions of S···C/C···S (4.0%) and S···S (2.0%) interactions were relatively low, they were still significant enough to confirm their presence in the crystal packing. In contrast, the contributions from C···N/N···C (0.5%) and C···C/C···C (0.3%) interactions were negligible, aligning with the findings of the X-ray structure.

Notably, the Hirshfeld surface analysis combined with the fingerprint plots revealed a complete absence of N···S/S···N interactions (0%). This finding elucidates the layered crystal packing observed in **DM3T–44BiPy**.

The comprehensive understanding of the atom pair interactions and their contributions provided by these analyses enhances our knowledge of the crystal packing of **DM3T–44BiPy**. The dominance of hydrogen-bonding interactions, along with the confirmation of other intermolecular contacts, explains the layered organization observed in the crystal structure. Further investigations can delve into the implications of these interactions on the physical and chemical properties of **DM3T–44BiPy**, facilitating the development of advanced materials and functional devices.

#### 2.3.3. DM3TMeOH–44BiPy

Single X-ray diffraction analysis confirmed the presence of both constituents, **DM3TMeOH** and **44BiPy**, in the cocrystal (Figure 9). The supramolecular complex belonged to the triclinic space group *P*$\bar{1}$, with the asymmetric unit consisting of one **DM3TMeOH** and one **44BiPy** molecule. In the structure, a cyclic motif was formed by four units, comprising two bipyridine and two terthiophene molecules (Figure 9). The organization of these units was governed by HBs involving C–H···O, O–H···N, and C–H···S interactions (Appendix A). Specifically, the hydrogen bonds O1–H1···N2 and C15–H15B···N1, with lengths of 2.053 Å and 2.557 Å, respectively, played a crucial role in forming the cyclic arrangement observed in the crystal structure. More details about the metrics of noncovalent interactions can be found in Table 3.

Remarkably, the presence of intramolecular hydrogen bonding influenced by C7–H7···O1 (with a length of 2.389 Å) led to the adoption of the S-*cis* conformation by the 5-methyl-2-thienyl unit attached to the 2-position of the central 3-thiophenemethanol unit in **DM3TMeOH**.

The findings from the X-ray diffraction analysis provide valuable insights into the structural characteristics and bonding arrangements within the **DM3TMeOH–44BiPy** cocrystal. Understanding the intermolecular and intramolecular hydrogen-bonding interactions contributes to our knowledge of the stability and conformational preferences of the cocrystal. These insights can guide further investigations into the properties and potential applications of this cocrystal system (see Section 2.4).

In the crystal structure, the arrangement of molecules in the **DM3TMeOH–44BiPy** cocrystal exhibited the formation of molecular sheets that are parallel to the *bc* plane (as depicted in Figure 10). These molecular sheets were constructed through weak hydrogen bonds between neighboring cyclic motifs, specifically, the hydrogen bond C8–H8···O1, with a bond length of 2.616 Å.

Furthermore, the molecules were held together by additional hydrogen bonds along the *a*-axis. These hydrogen bonds included C15–H15A···S1, with a bond length of 2.950 Å, and C15–H15A···π, with a separation measured at 2.638 Å. The C–H···π interactions coupled with the intermolecular HBs played a significant role in stabilizing the crystal structure of the **DM3TMeOH–44BiPy** cocrystal (Figure 11).

Like **DM3T–44BiPy** cocrystal, the Hirshfeld surface analysis of **DM3TMeOH–44BiPy** revealed the presence of red spots corresponding to various hydrogen bonds observed in the X-ray structure (Figure 12). Notably, due to the hydroxymethyl (–CH_2_OH) group present in **DM3TMeOH**, the red spot corresponding to the O···H interaction is particularly prominent on the surface, indicating its strength and size.

The relative contributions of different interactions to the overall surface are as follows: H···H (38.4%), C···H/H···C (32.3%), S···H/H···S (14.1%), N···H/H···N (7.2%), O···H/H···O (3.4%), S···C/C···S (2.6%), S···S (1.4%), and C···C/C···C (0.8%) (Figure 13). Once again, the major contributions arose from the simultaneous occurrence of various hydrogen-bonding interactions. However, an important observation is that the relative proportions of N···H/H···N (7.2%) and S···H/H···S (14.1%) interactions in **DM3TMeOH–44BiPy** were reversed compared to those in **DM3T–44BiPy**, where N···H/H···N accounted for 12.9% and S···H/H···S accounted for 9.9%. This reversal was attributed to the extensive complex network of hydrogen bonds observed within the **44BiPy** layer of the **DM3T–44BiPy** crystal, which changed to linear hydrogen-bonding interactions between **DM3TMeOH** and **44BiPy** in the **DM3TMeOH–44BiPy** cocrystal.

Furthermore, the proportion of S···S (1.4%) interactions confirms their presence in the crystal packing, while the insignificant proportion of C···C/C···C (0.8%) interactions ruled out the occurrence of π–π stacking, which aligns with the observations from the X-ray structure of **DM3TMeOH–44BiPy**.

### 2.4. Photophysical Properties

In the literature, the photophysical properties of α-oligothiophenes (n = 1–7, where n is the number of thiophene rings) have been extensively studied both theoretically and experimentally [28,29]. We also investigated the photophysical properties of our newly synthesized terthiophene derivatives, **DM3T** and **DM3TMeOH**, and their binding with **44BiPy** in solution and solid-state using UV/visible spectroscopy and fluorescence spectroscopy. The UV/visible spectra of all materials were recorded in chloroform solution at room temperature and are presented in Appendix A.

As anticipated, the UV/visible absorption spectrum of **DM3T** in chloroform solution was devoid of structural features and exhibited a single broad absorption band at 365 nm (reported literature value of λ*_abs,max_* at 354 nm in dioxane [28]). This characteristic electronic spectrum is typical of oligothiophenes lacking distinct structural features and is attributed to the π(1^1^A_g_)–π*(1^1^B_u_) transition (S_0_→S_1_) of the oligothiophene backbone [28,30]. Importantly, this electronic spectrum is known to be unaffected by the presence of foreign species; in line with expectations, the presence of **44BiPy** did not show any noticeable effect on the absorption spectra of **DM3T** [30]. In the solid state, the overall absorption spectrum region remains unaffected but the absorbance of both **DM3T** and **DM3T–44BiPy** decreased significantly as the bulk of the material remained unexcited.

Similarly, **DM3TMeOH** was also devoid of structural features and displayed only a single broad absorption band with slight blue-shift by 8 nm compared to that of **DM3T**. This blue-shift was attributed to the *S*-cis configuration of the terthiophene units in the ground state. Like **DM3T** in the solution state, the absorption spectrum of **DM3TMeOH** remained unaffected by the presence of **44BiPy**. In the solid state, **DM3TMeOH–44BiPy** exhibited similar absorption to that of **DM3TMeOH** in the solution state.

Moving on to the emission properties, Figure 14 depicts the fluorescence spectra of all materials, recorded both in solution and in the solid state at room temperature along with emission colors under a UV lamp (λ = 365 nm). The fluorescence spectrum exhibits the distinctive vibronic features typically observed in terthiophenes. The origin of these vibronic features in the fluorescence spectrum of oligothiophenes has been the subject of detailed theoretical and experimental investigations, which have been reported in previous studies [28,29,30,31]. It is well established that the α-oligothiophenes adopt a quinoid planar geometry in the excited state. As a result of rigid and planar structure, the fluorescence spectra appeared with well-resolved vibronic features as the molecules of α-oligothiophenes relaxed back to the aromatic ground state radiatively.

In chloroform solution, **DM3T** exhibited characteristic well-resolved vibronic features at 421 nm, 442 nm, and 473 nm. This is consistent with the previous literature that reported maximum fluorescence at 407 nm and 426 nm in dioxane [28]. Compared to the solution state, the entire fluorescence spectrum of **DM3T** experienced a red-shift of 40 nm in the solid state due to the aggregation-caused quenching (ACQ) effect [32], while maintaining its structural features (Figure 14a,b).

Upon the addition of **44BiPy**, fluorescence quenching was observed in the solution state, and this quenching became more pronounced in the solid state. This suggests an electron transfer process from **DM3T** to **44BiPy** in the excited state. Interestingly, in the solid state of **DM3T–44BiPy**, the vibronic band at 465 nm completely disappeared, and the broad shoulder of **DM3T**’s emission spectrum became a distinct emission band. On the basis of the energetics of **DM3T**’s electronic states, this indicates that electron transfer occurred from **DM3T**’s 1*B*_u_ electronic state to the lowest unoccupied molecular orbital (LUMO) of 44BiPy.

Similarly, **DM3TMeOH** exhibited vibronic bands at 428 nm and 448 nm in its fluorescence spectrum in the solution state. In the presence of **44BiPy**, pronounced fluorescence quenching was observed due to the formation of strong hydrogen bonds (Figure 14c). In the solid state, both **DM3TMeOH** and **DM3TMeOH–44BiPy** became non-luminescent due to aggregation driven by strong hydrogen bonding (Figure 14d).

## 3. Discussion

In this section, we provide a comparative discussion of the X-ray structure and the Hirshfeld surface analysis of **DM3T**, **DM3T–44BiPy**, and **DM3TMeOH–44BiPy**. The crystallographic data and refinement parameters for these compounds are summarized in Appendix A. **DM3T** crystallized in the monoclinic crystal system with space group *P*2_1_, while both **DM3T–44BiPy** and **DM3TMeOH–44BiPy** crystallized in the triclinic space group *P*$\bar{1}$.

In the X-ray structure of **DM3T**, all thiophene units adopted an *S-trans* conformation with a slightly curved molecular plane geometry. This conformation was maintained within the **DM3T** layer of its cocrystal with **44BiPy**, although the planar geometry was further distorted. In contrast, the thiophene units in **DM3TMeOH** adopted a combination of *S-trans* and *S-cis* conformations. The *S-trans* portion of the molecule remained planar, while the *S-cis* part adopted an out-of-plane geometry.

The **DM3T** molecules in the **DM3T–44BiPy** cocrystal continued to exhibit a herringbone pattern. However, due to the absence of desired N···S interactions, the strong hydrogen bonds within **DM3T** and **44BiPy** molecules resulted in an almost independent layered structure of **DM3T–44BiPy**. On the other hand, the presence of a hydroxymethyl group on **DM3TMeOH** ensures that the self-assembly of **DM3TMeOH** with **44BiPy** was exclusively directed by hydrogen bonding involving O–H···N interactions, forming a cyclic motif.

The resolved 2D fingerprint plots in all cases revealed that hydrogen bonds were the major contributors to the overall Hirshfeld surface. Interestingly, the relative proportions of N···H/H···N (12.9%) and S···H/H···S (9.9%) interactions in **DM3T–44BiPy** were reversed compared to N···H/H···N (7.2%) and S···H/H···S (14.1%) interactions in **DM3TMeOH–44BiPy**. This change in proportions reflected the difference in crystal packing, transitioning from an almost independent layered cocrystal to a molecular packing arrangement involving a cyclic motif, although both self-assemblies were directed by hydrogen bonds.

In summary, the X-ray structures and Hirshfeld surface analyses provided valuable insights into the molecular arrangements and intermolecular interactions in **DM3T**, **DM3T–44BiPy**, and **DM3TMeOH–44BiPy**. The differences observed in their crystal packings and hydrogen-bonding patterns highlighted the influence of molecular composition and functional groups on the supramolecular organization of these cocrystals. The findings of photophysical properties investigations demonstrated the influence of **44BiPy** on the fluorescence properties of both **DM3T** and **DM3TMeOH**, suggesting electron transfer and fluorescence-quenching effects in the presence of **44BiPy**. The solid-state behavior of these systems highlighted the impact of aggregation and hydrogen bonding on their luminescent properties.

## 4. Materials and Methods

General experimental information. Solution NMR spectroscopy analyses were performed at the Centre d’Instrumentation en Resonance Magnétique (CIREM) of the Université Libre de Bruxelles in Belgium. ^1^H- and ^13^C-NMR spectra were recorded on JEOL 400 Hz spectrometers at 298 K. CDCl_3_ was used as received (purchased from Eurisotop, France). The residual nondeuterated solvent was used as an internal standard for calibration of ^1^H- and ^13^C-NMR spectra. Chemical shifts (δ) are expressed in ppm, and coupling constants values (^n^J) are expressed in Hz. Abbreviations used for NMR spectra are as follows: s, singlet; d, doublet; m, multiplet.

Materials. 2,5-Dibromothiophene (98%, CAS registry no. 3141-27-3) was purchased from BLD Pharmatech GmbH (Kaiserslautern, Germany). 5-Methylthiophene-2-boronic acid pinacol ester (95%, CAS registry no. 476004-80-5), potassium carbonate anhydrous (>99%, CAS registry no. 584-08-7), and toluene (>99%, CAS registry no.108-88-3) were bought from Merck (Darmstadt, Germany). 3-Thiophenemethanol (98%, CAS registry no. 71637-34-8), tetrabutylammonium fluoride (TBAF) solution (1.0 M in THF, CAS no. 429-41-4), and N-bromosuccinimide (NBS) (ReagentPlus^®^, 99%, CAS no. 128-08-5) were purchased from Sigma-Aldrich (now known as Merck) (Hoeilaart, Belgium). Tetrakis(triphenylphosphine)palladium (0) (99%, CAS registry no. 14221-01-3) was obtained from Apollo Scientific (Cheshire, UK). 4,4′-Bipyridine (>98%, CAS registry no. 553-26-4) was bought from TCI Chemicals (Zwijndrecht, Belgium). Ethanol absolute (>99.8%, CAS registry no. 64-17-5) was obtained from VWR Chemicals (Leuven, Belgium). Dichloromethane (>98%, CAS registry no. 75-09-2) and N,N-dimethylformamide (>99.5%, CAS registry no. 68-12-2) were purchased from Chem-Lab NV, (Zedelgem, Belgium). All materials were used as received. The air-sensitive reaction was performed using standard Schlenk techniques under an argon atmosphere. Flash column chromatography was carried out using silica gel (60 Å, 70–200 μm) purchased from DAVISIL-Grace GmbH, Germany. Analytical thin-layer chromatography was performed using silica plates with aluminum backings (250 μm with F-254 indicator, Merck, Germany), and the results were visualized using a 254/365 nm UV lamp.

Synthesis 5,5″-dimethyl-2,2′:5′,2″-terthiophene (**DM3T**). A mixture of 2,5-dibromothiophene (200 mg, 0.83 mmol, 1 equiv.), 5-methylthiophene-2-boronic acid pinacol ester (390 mg, 0.414 mL, 1.74 mmol, 2.1 equiv.), and Pd(PPh_3_)_4_ (95 mg, 10 mol.%) was dissolved in toluene (10 mL, bubbled with argon gas for 15 min) and heated to 90 °C. A solution of K_2_CO_3_ (2 M, 1.25 mL, EtOH/H_2_O (1:1), bubbled with argon gas for 5 min) was then added to the reaction mixture and stirred at the same temperature for 24 h. After cooling to room temperature, the reaction mixture was diluted with ethyl acetate and washed with water followed by brine solution. The organic layer was dried over anhydrous sodium sulfate and filtered. Solvent was removed and residue was purified by silica gel column chromatography using ethyl acetate/hexane (v/v = 1:4) as an eluent to afford the title compound as a bright-yellow solid (198 mg; yield 86%). ^1^H-NMR (400 MHz, CDCl_3_) δ 7.00–6.88 (m, 4H), 6.65 (d, J = 0.9 Hz, 2H), 2.47 (s, 6H) (Appendix A). ^13^C-NMR (101 MHz, CDCl_3_) δ 139.25, 136.13, 135.00, 126.04, 123.58, 123.49, 15.44 (Appendix A).

Synthesis of 2,5-dibromo-3-thiophenenemethanol (**DBTMeOH**). The title compound was synthesized according to a slightly modified literature procedure [33]. To a solution of 3-thiophenemethanol (1.7 g, 14.9 mmol, 1 equiv.) in dichloromethane (50 mL) at 0 °C was added NBS (5.89 g, 32.7 mmol, 2.2 equiv.) in DMF (10 mL) dropwise. The reaction mixture was stirred at room temperature overnight. The reaction mixture was quenched by addition of water and diluted with dichloromethane. The organic layer was washed with brine, dried over anhydrous sodium sulfate, and concentrated. The crude product was purified by silica gel column chromatography using hexane/ethyl acetate (80:20) as an eluent to afford the title compound as a white solid (3.45 g, 85% yield). ^1^H-NMR (400 MHz, CDCl_3_) δ 6.99 (s, 1H), 4.52 (s, 2H). ^13^C-NMR (101 MHz, CDCl_3_) δ 141.53, 130.61, 111.53, 109.33, 59.37.

Synthesis of tert-butyl((2,5-dibromothiophen-3-yl)methoxy)dimethylsilane (**DBTMeOTBDMS**). The title compound was synthesized according to a literature procedure [34]. A mixture of **DBTMeOH** (3.34 g, 12.2 mmol, 1 equiv.), *tert*-butyldimethylsilyl chloride (2.78 g, 18.4 mmol, 1.5 equiv.), and imidazole (1.67 g, 24.5 mmol, 2 equiv.) was dissolved in DMF (10 mL) and heated at 65 °C for 2 h. After cooling to room temperature, the reaction mixture was diluted with ethyl acetate and washed with water. The organic layer was washed with brine, dried over anhydrous sodium sulfate, and concentrated. The crude product was purified by silica gel column chromatography using hexane as eluent to afford the title compound as a colorless liquid (4.88 g, quantitative yield). ^1^H-NMR (400 MHz, CDCl_3_) δ 6.96 (s, 1H), 4.54 (s, 2H), 0.91 (d, J = 0.8 Hz, 9H), 0.08 (d, J = 0.8 Hz, 6H).

Synthesis of tert-butyl((5,5″-dimethyl-[2,2′:5′,2″-terthiophen]-3′-yl)methoxy)dimethylsilane (**DM3TMeOTBDMS**). The title compound was prepared using the well-known palladium-catalyzed Suzuki cross-coupling reaction. A mixture of **DBTMeOTBDMS** (1.0 g, 4.13 mmol, 1 equiv.), 5-methylthiophene-2-boronic acid pinacol ester (2.4 g, 2.56 mL, 9.09 mmol, 2.2 equiv.), and Pd(PPh_3_)_4_ (300 mg, 5 mol.%) was dissolved in toluene (20 mL, bubbled with argon gas for 15 min) and heated to 90 °C. A solution of K_2_CO_3_ (2 M, 2 mL, EtOH/H_2_O (1:1), bubbled with argon gas for 5 min) was then added to the reaction mixture and stirred at the same temperature for 24 h. After cooling to room temperature, the reaction mixture was diluted with ethyl acetate and washed with water, followed by brine solution. The organic layer was dried over anhydrous sodium sulfate and filtered. The solvent was removed, and the residue was purified by silica gel column chromatography using hexane/dichloromethane (85:15) as an eluent to afford the title compound as a light-green solid (0.88 g; 77% yield). ^1^H-NMR (400 MHz, CDCl_3_) δ 7.10 (s, 1H), 6.92 (dd, J = 7.8, 3.5 Hz, 2H), 6.72–6.69 (m, 1H), 6.66–6.63 (m, 1H), 4.73 (s, 2H), 2.49 (d, J = 1.0 Hz, 3H), 2.47 (d, J = 1.0 Hz, 3H), 0.92 (s, 9H), 0.08 (s, 6H). ^13^C-NMR (101 MHz, CDCl_3_) δ 140.49, 139.26, 138.57, 135.31, 135.02, 133.13, 130.25, 126.08, 126.02, 125.96, 125.09, 123.50, 60.04, 26.02, 18.46, 15.44, 15.38, −5.12.

Synthesis of (5,5″-dimethyl-[2,2′:5′,2″-terthiophen]-3′-yl)methanol (**DM3TMeOH**). To a solution of **DM3TMeOTBDMS** (0.88 g, 2.09 mmol, 1 equiv.) in THF (10 mL) at room temperature was added tetrabutylammonium fluoride (2.5 mL, 1 M solution in THF, 1.2 equiv.) and stirred. After 1 h, THF was removed under reduced pressure, and the residue was dissolved in dichloromethane. The organic layer was washed with brine, dried over anhydrous sodium sulfate, and concentrated. The crude product was absorbed on silica gel and eluted with dichloromethane to afford the title compound as green solid (0.7 g, quantitative yield). ^1^H-NMR (400 MHz, CDCl_3_) δ 7.11 (s, 1H), 6.97 (d, J = 3.5 Hz, 1H), 6.95 (d, J = 3.5 Hz, 1H), 6.74–6.70 (m, 1H), 6.68–6.64 (m, 1H), 4.72 (s, 2H), 2.50 (s, 3H), 2.48 (s, 3H) (Appendix A). ^13^C NMR (101 MHz, CDCl_3_) δ 140.93, 139.56, 137.61, 136.01, 134.63, 132.55, 132.18, 126.48, 126.11, 126.09, 125.08, 123.74, 59.23, 15.46, 15.38 (Appendix A).

Data collection and refinement. Single-crystal data collection (at 100 K) and evaluation of the cocrystal, **DM3T–44BiPy**, was performed on a Rigaku Oxford Diffraction Supernova Dual Source (Cu at zero) diffractometer equipped with an Atlas CCD detector using ω scans and Mo Kα (λ = 0.71073 Å) radiation. The images were interpreted and integrated with the program CrysAlisPro [35]. Using Olex2 [36], the structure was solved by direct methods using the ShelXT structure solution program and refined by full-matrix least squares on F2 using the ShelXL program package [37,38]. Non-hydrogen atoms were anisotropically refined, and the hydrogen atoms were in riding mode with isotropic temperature factors fixed at 1.2 times U(eq) of the parent atoms.

Cambridge structural database survey. A simple search was launched using the string “terthiophene” in the compound name option. The search revealed more than 200 crystal structures including quaterthiophene. The hitlist was manually screened for 2,2′:5′,2”-terthiophene (**3T**) with or without terminal substituents only and the results were reduced to following 23 CCDC entries: 112398, 232984, 254856, 619628, 644333, 644334, 646597, 699323, 747221, 800037, 836023, 848505, 1005911, 1040724, 1040725, 1221168, 1862144, 1991924, 1991925, 2044752, 2062986, 19862144, and 1981015. Apart from these homo-crystal entries, only one crystal structure is reported as a two-component cocrystal of **3T** with 7,7,8,8-tetracyanoquinodimethane (TCNQ) (CCDC 1862143), and only one crystal structure is reported as a three-component cocrystal consisting of **3T**, guanidinium, and 4,4’-azobenzenedisulfonate (CCDC 797309).

UV/visible and fluorescence spectroscopy. The HPLC-grade chloroform solvent was used for solution preparations. All solutions were in ranges of 10^−5^ to 10^−6^ M. Solid-state samples were prepared using the solution drop-cast method on a glass slide by dissolving 1 mg of sample in 100 µL of chloroform. Absorption spectra were recorded on a Shimazu spectrophotometer UV-1800, and fluorescence spectra were recorded on a HORIBA Fluoromax-4_1507D-3516-FM fluorescence spectrometer at room temperature. For **DM3T** and **DM3T–44BiPy**, the excitation wavelength was 365 nm for both the solution and the solid state, while the excitation wavelength of 357 nm was used for **DM3TMeOH** and **DM3TMeOH–44BiPy** samples.

## 5. Conclusions

In our study, we presented the synthesis and self-organization of methyl-capped terthiophene (**DM3T**) and its methyl alcohol derivative in combination with 4,4′-bipyridine (**44BiPy**). Through single-crystal X-ray diffraction analysis, we made the unexpected discovery of a three-dimensional stacked, layered organization of **DM3T** and **44BiPy**. These layers were found to be nearly independent of each other and held together by weak hydrogen bonds. The investigation of the photophysical properties of this system revealed a strong fluorescence quenching of **DM3T** by **44BiPy** in the solid state. This suggests an efficient charge transfer between the two components. Such phenomena are of significant interest in the development of energy-generation devices, particularly in the field of organic solar cells. Furthermore, we observed that **DM3TMeOH**, the methyl alcohol derivative of **DM3T**, organized itself in a closed cyclic motif with 44BiPy through strong hydrogen bonds. Overall, our findings shed light on the unique self-organization and supramolecular arrangements of **DM3T** and its derivatives with **44BiPy**. The unexpected 3D stacked, layered organization, along with the efficient charge transfer offers new avenues for future research in self-organized layered materials. Further exploration of the photophysical properties and applications of this system could lead to advancements in the development of energy generation devices.

## Data Availability

The data presented in this study are available in the Appendix A.

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
