# Peer review of "Insight into Unusual Supramolecular Self-Assemblies of Terthiophenes Directed by Weak Hydrogen Bonding"

_ijms, 2023, doi:10.3390/ijms241311127_

Round 1

Reviewer 1 Report

In this manuscript, the authors reported the self-assembly of terthiophene and its methyl alcohol derivative with 4,4’-bipyridine. the single crystal X-ray diffraction analysis revealed that DM3T and 44BiPy consist of stacked, almost independent, infinite 2D layers while held together by weak hydrogen bonds. Then, the photophysical properties were studied. I recommend the paper to published after minor revision

1- The resolution and content of Figure 14 are not clear; the authors need to improve.

2- The UV-Visible spectroscopy of DM3T, DM3TMeOH, and their binding with 44BiPy in the solution and solid-state need to provide.

3- All  PL spectra have two overlapped peaks. why?

4- The photophysical properties need to be more clearly discussed, can see the following papers as references:  https://doi.org/10.1016/j.polymer.2018.09.052

and DOI: 10.1039/C8RA02369G 

Good

Author Response

In this manuscript, the authors reported the self-assembly of terthiophene and its methyl alcohol derivative with 4,4’-bipyridine. the single crystal X-ray diffraction analysis revealed that DM3T and 44BiPy consist of stacked, almost independent, infinite 2D layers while held together by weak hydrogen bonds. Then, the photophysical properties were studied. I recommend the paper to be published after minor revision.

1- The resolution and content of Figure 14 are not clear; the authors need to improve.

Resolution of Figure 14 has been improved in revised manuscript.

2- The UV-Visible spectroscopy of DM3T, DM3TMeOH, and their binding with 44BiPy in the solution and solid-state need to provide.

UV-visible spectra of DM3T, DM3T-44BiPy, DM3TMeOH, and DM3TMeOH-44BiPy included in supporting information document.

3- All  PL spectra have two overlapped peaks. why?

In the excited state, terthiophene adopts a quinoid planar geometry. As a result of rigidity and planar structure, the fluorescence spectra appear with well-resolved vibronic features as the molecules relax back to aromatic ground state radiatively. The same has been elaborated in the revised manuscript.

4- The photophysical properties need to be more clearly discussed, can see the following papers as references:  https://doi.org/10.1016/j.polymer.2018.09.052

and DOI: 10.1039/C8RA02369G

The section describing the photophysical properties has been modified and rewritten comprehensively in the revised manuscript.

Reviewer 2 Report

In this work, the authors demonstrated a novel 2D layered self-organization pattern of terthiophene DM3T and its methyl alcohol derivative with 44BiPy, along with a detailed structural and fluorescence characterization of the DM3T based compounds. This paper provides valuable insights into the structural characteristics of DM3T based (co-) crystals and the resulting impact on the photophysical properties, and thus is suitable for the Int. J. Mol. Sci. journal publication. Nevertheless, there are a few points left in the current manuscript to be discussed further, and thus I would recommend minor visions before publishing:

·        Self-organization structure in DM3T-44BiPy structure is the most important observation in this work, and it’s been shown in the X-ray data (line 288) that there are two DM3T and three 44BiPy molecules in the asymmetric unit. It’s quite interesting that this ratio is the same as the initial ratio in the solution (line 222). Does the co-crystalized structure depend on the initial DM3T vs 44BiPy ratio? Why the ratio does not change during the crystallization process (for example phase segregation)? Are they 100% soluble in each other?

·        Following the last question – what’s the ratio of DM3TMeOH vs 44BiPy in the solution before crystallization? Is it 1:1, the same as what’s in the final single crystal?

·        At line 269 the authors stated that “The significant contribution from H···H interactions suggests the presence of both intermolecular as well as intramolecular hydrogen bonds within the crystal structure”, why is that? Could the authors elaborate more about why both inter- and intra- molecular hydrogen bonds can be referred from the high H···H interaction contributions?

·        There are multiple references in the paper showing “Error! Reference source not found…”, please fix them: line 208, line 236, line 305, line 306.

Author Response

In this work, the authors demonstrated a novel 2D layered self-organization pattern of terthiophene DM3T and its methyl alcohol derivative with 44BiPy, along with a detailed structural and fluorescence characterization of the DM3T based compounds. This paper provides valuable insights into the structural characteristics of DM3T based (co-) crystals and the resulting impact on the photophysical properties, and thus is suitable for the Int. J. Mol. Sci. journal publication. Nevertheless, there are a few points left in the current manuscript to be discussed further, and thus I would recommend minor visions before publishing:

  • Self-organization structure in DM3T-44BiPy structure is the most important observation in this work, and it’s been shown in the X-ray data (line 288) that there are two DM3T and three 44BiPy molecules in the asymmetric unit. It’s quite interesting that this ratio is the same as the initial ratio in the solution (line 222). Does the co-crystalized structure depend on the initial DM3T vs 44BiPy ratio? Why the ratio does not change during the crystallization process (for example phase segregation)? Are they 100% soluble in each other?

This is a very good question raised by the reviewer. We thank the reviewer for this comment. In the present study, we did not verify whether the co-crystal composition depends on the initial ratios of DM3T and 44BiPy or not. In the case of the DM3T-44BiPy co-crystal structure which has the same composition as the initial ratios of its constituents could be purely coincidental. However, the dependency of composition of co-crystal on the initial ratios of DM3T and 44BiPy cannot be ruled out. Therefore, authors mutually agreed upon taking up this task and disseminate the results as an independent study if there is evidence of such dependency.

For crystallization, a dichloromethane (DCM)/hexane solvent mixture was chosen. Both DM3T and 44BiPy are readily soluble in DCM and almost insoluble in hexane.

  • Following the last question – what’s the ratio of DM3TMeOH vs 44BiPy in the solution before crystallization? Is it 1:1, the same as what’s in the final single crystal?

The ratio of DM3TMeOH and 44BiPy was 2:1 in solution and found to be the same in final single co-crystal. The same has been mentioned in the revised manuscript.

  • At line 269 the authors stated that “The significant contribution from H···H interactions suggests the presence of both intermolecular as well as intramolecular hydrogen bonds within the crystal structure”, why is that? Could the authors elaborate more about why both inter- and intra- molecular hydrogen bonds can be referred from the high H···H interaction contributions?

This statement was technically incorrect and has been corrected in the revised manuscript. Now, the statement reads as follows:  

“The significant contribution from H···H interactions suggests that the vast majority of London dispersion forces are acting within the crystal structure.”

  • There are multiple references in the paper showing “Error! Reference source not found…”, please fix them: line 208, line 236, line 305, line 306.

We did not find any issue with our version. It probably comes from the word to pdf conversion of the reviewer. We had a problem in a preliminary version but it was fixed.

Reviewer 3 Report

Before publication, the authors are invited to consider the following suggestions and remarks:

1.      One suggestion is regarding the title length. Can you create a shorter and more    comprehensive one?

2.      In the supplementary materials part, it would be better to include the 1H NMR spectrum of the synthesized terthiophene compounds. 

3.      It would be nice if you can insert, in figure 14, a photograph with the color emission of the solutions and solid state of the investigated terthiophene derivatives under UV lamp.

Author Response

Before publication, the authors are invited to consider the following suggestions and remarks:

  1. One suggestion is regarding the title length. Can you create a shorter and more comprehensive one?

We have shortened the title and new title reads as follows:

“Insight into unusual supramolecular self-assemblies of terthiophenes directed by hydrogen bonding”

  1. In the supplementary materials part, it would be better to include the 1H NMR spectrum of the synthesized terthiophene compounds.

1H and 13C NMR spectra of synthesized terthiophenes have been included in supplementary materials document.

  1. It would be nice if you can insert, in figure 14, a photograph with the color emission of the solutions and solid state of the investigated terthiophene derivatives under UV lamp.

Photographs displaying emission color of investigated terthiophene under UV lamp have been inserted in Figure 14.